# The Diagnosis and Treatment of Fungal Endophthalmitis: An Update

**DOI:** 10.3390/diagnostics12030679

**Published:** 2022-03-10

**Authors:** Ciprian Danielescu, Horia Tudor Stanca, Raluca-Eugenia Iorga, Diana-Maria Darabus, Vasile Potop

**Affiliations:** 1Department of Ophthalmology, Faculty of Medicine, University of Medicine and Pharmacy “Grigore T. Popa”, Strada Universitatii No. 16, 700115 Iasi, Romania; ciprian.danielescu@umfiasi.ro; 2Department of Ophthalmology, Faculty of Medicine, University of Medicine and Pharmacy “Carol Davila”, Strada Dionisie Lupu No. 37, 020021 București, Romania; horia.stanca@umfcd.ro (H.T.S.); vasile.potop@umfcd.ro (V.P.); 3Department of Ophthalmology, Faculty of Medicine, University of Medicine and Pharmacy “Victor Babes”, Piata Eftimie Murgu No. 2, 300041 Timisoara, Romania

**Keywords:** fungal endophthalmitis, polymerase chain reaction, next generation sequencing, intravitreal injection, pars plana vitrectomy

## Abstract

In recent, large case series of fungal endophthalmitis (FE) that were published by Asian authors, the most frequent etiologic agents for all types of FE are molds (usually *Aspergillus* species, while *Fusarium* is the prevalent etiology in keratitis-related FE). *Candida* was the organism found in most cases of endogenous FE. However, we must keep in mind that prevalence of fungal species varies with the geographical area. Lately, polymerase chain reaction (PCR) was increasingly used for the diagnosis of FE, allowing for very high diagnostic sensitivity, while the costs become more affordable with time. The most important shortcoming of PCR—the limited number of pathogens that can be simultaneously searched for—may be overcome by newer techniques, such as next-generation sequencing. There are even hopes of searching for genetic sequences that codify resistance to antifungals. We must not forget the potential of simpler tests (such as galactomannan and β-d-glucan) in orienting towards a diagnosis of FE. There are few reports about the use of newer antifungals in FE. Echinocandins have low penetration in the vitreous cavity, and may be of use in cases of fungal chorioretinitis (without vitritis), or injected intravitreally as an off-label, salvage therapy.

## 1. Introduction

Endophthalmitis is a serious ophthalmic condition, carrying the risk of permanent visual loss. Knowledge of its diagnosis and treatment is of essence for every ophthalmologist. In endophthalmitis the internal structures of the eye are invaded by replicating microorganisms, resulting in an inflammatory response [1]. The term endophthalmitis is usually reserved for bacterial or fungal infections, while inflammation of viral or parasitical cause is considered a form of uveitis. The causative organism may be directly inoculated into the eye (exogenous endophthalmitis, usually posttraumatic or post intraocular surgery) or may enter through hematogenous spread from distant foci (endogenous endophthalmitis). In fungal endophthalmitis (FE), the causative organism is either a mold or yeast.

The diagnosis and treatment of FE are challenging due to a series of particularities, especially the difficult, time consuming etiological diagnosis and the problems of antifungal therapy (availability, efficacy, potential toxicity). The relative rarity of the FE has led to the fact that there is no level 1 evidence to guide its management [2]. This is why our most useful information regarding this matter comes from case series (some of which, recently published, are very large) [3].

This review aims to provide the reader with the latest published information regarding FE, focusing on recent diagnostic techniques and on the advances in the use of antifungal drugs in ophthalmology.

## 2. Epidemiology

Several authors have stated that FE has increased from 8.6% to 18.6% of culture-positive cases over the last 20 years [4,5,6]. (Table 1) Due to the characteristics of fungal growth, acute postoperative FE is considered a rarity, but fungi are considered responsible for 16–27% of cases of delayed-onset postoperative endophthalmitis [7].

In the largest retrospective study of patients with fungal endophthalmitis, which took place in India, 46.8% were postoperative cases, 35.6% posttraumatic and 17.5% endogenous [3]. Another large study reported that 6.5% of post-traumatic endophthalmitis cases had a fungal etiology (16.7% of culture-proven cases) [8]. Endogenous FE has the particularity of being bilateral in many cases (27.2% of patients in a review) [9].

In case series of endogenous endophthalmitis published in the last 5 years, 6% to 50% were of fungal origin [9,11,12,13,14,15,16], while 59% of intravenous drug-abuse related cases had fungal etiology [17]. In a USA national retrospective cross-study, 13.7% of patients with endogenous endophthalmitis had a history of drug dependence. While the authors did not provide information regarding the etiology, it was noted that 3.1% of all cases (and 4.6% of the drug-using patients) had a diagnosis of fungemia [19].

### Predisposing Medical Conditions

In several series of patients with endogenous FE, malignancies were a preexisting condition in 21.45 to 69.7% of cases. Intravenous drug abuse was also a frequent risk factor, present in 15.4% to 28.6% of fungal FE [9]. Patients who have recently suffered a general surgery were also at risk (28.6% to 37.9% of all endogenous FE cases) [20,21]. Subjects diagnosed with FE caused by mold species had in higher proportions a history of iatrogenic immunosuppression, whole-organ transplantation, or an indwelling venous line [20,22].

## 3. Etiology and Pathogeny

In the largest retrospective study of patients with FE, which took place in India, 39.0% were caused by *Aspergillus* species, 15.1% by *Candida* species, 15.9% by *Fusarium* species, and 30.0% by other fungi [3]. Another large retrospective review of endophthalmitis cases from North India also found that *Aspergillus* was the predominant cause of FE (36.1%), followed by *Fusarium* (26.4%) [23]. A classic scenario is an ocular trauma that took place in a rural setting, perhaps involving contamination with soil or vegetal matter.

In a recent review of 59 articles, the most common etiology of post-cataract FE was *Aspergillus*, followed by *Fusarium*. [6].

In a retrospective study of patients from South India diagnosed with *Aspergillus* endophthalmitis, 46% of cases were associated with penetrating trauma, 33% were acute -onset postoperative cases, 6.5% delayed-onset postoperative cases and 11% endogenous [24]. *Aspergillus flavus* was the commonest infecting species. Also, in retrospective series of posttraumatic fungal endophthalmitis (patients from eastern China), 74.3% of cultures have grown *Aspergillus* species [25].

In a large series of patients with fungal keratitis, 9.4% developed an endophthalmitis. The risk factors for endophthalmitis were: topical steroid use, previous corneal laceration suturing, large corneal ulcer size (≥10 mm diameter), hypopyon and aphakia, while the advent of corneal perforation was not a significant risk factor. The most frequent etiologies were *Fusarium* (40.5%) and *Aspergillus* (16.2%) [26]. In another series of patients from northern China, 73.3% of FE associated with fungal keratitis were caused by *Fusarium* species [27].

Endophthalmitis after intravitreal injections is a very rare occurrence (0.02 to 0.5%) [2], and fungal etiology appears to be extremely rare [28]. There were reports of outbursts of FE following intravitreal injections contaminated with *Bipolaris hawaiiensis* [29,30], but recently there are fewer and fewer cases like that, probably due to the fact that ophthalmologists have performed less compounding pharmacy-prepared intravitreal injections.

Recently, there have been a few reports of cases of FE in patients hospitalized for COVID-19 pneumonia [31,32,33]. In a retrospective report of 24 patients from India, diagnosed with COVID-19 and endogenous endophthalmitis, 78.5% were of fungal etiology [31].

In a review of several series of patients with endogenous FE, the predominant microorganisms were yeasts (71.4% to 76.1%), most frequently *Candida* species (50% to 65%), while *Aspergillus* was the most frequent mold (in 11.7% to 16.4% of cases) [9]. However, *Aspergillus* was the most prevalent (29.7%) and *Candida* species followed closely (26.6%) in the largest retrospective study from India [3]. In cases of infantile endogenous endophthalmitis, *Candida* species have been characterized as the primary responsible organisms in multiple case series and reports in the United States [34]. The most common systemic risk factors were: prematurity, respiratory disorders, intraventricular hemorrhage, birth trauma, necrotizing enterocolitis, intrauterine hypoxia and birth asphyxia.

The pathogeny of endogenous endophthalmitis is hematogenous dissemination from distant foci; it affects primarily the choroidal space, due to the comparatively large blood flow, and then it spreads into the retina and vitreous [3].

In histopathology studies, *Candida* species seem to sequester preferentially in inflammatory nodules (another explanation for the fact that negative cultures should be interpreted with caution) [35]. There is, however, a clinicopathologic study of enucleated eyes with endogenous FE where the primary focus of infection with *Candida* was the vitreous; whereas subretinal or sub-retinal pigment epithelium infection (with invasion of retinal and choroidal vessel walls) was noted in eyes with aspergillosis [36]. In a murine model of fungal endophthalmitis, the infected retina exhibited induction of inflammatory mediators (TNFα, IL-1β and IL 6) with increased polymorphonuclear neutrophil infiltration. Histological analysis revealed heavy cellular infiltrates in the vitreous cavity, disruption of normal retinal architecture and retinal cell death [37].

## 4. Diagnosis

### 4.1. Clinical Diagnosis

In most cases FE does not have an acute clinical presentation. In one study, the mean latent periods were 7 days for post-traumatic FE, 20 days for postoperative and 30 days for endogenous FE [38]. In a retrospective series of posttraumatic FE (patients from eastern China), the time from trauma to the diagnosis of endophthalmitis was 2–4 weeks in 37.1% of patients and over 4 weeks in 42.9% [25]. Delayed diagnosis or initial misdiagnosis was reported in 16% to 63% of cases [39], some cases being initially treated as non-infectious uveitis. Eyelid edema, conjunctival injection, anterior chamber cells, flare or hypopyon, vitreous inflammation and chorioretinitis are frequent but non-specific signs. Focal vitreous opacities (“string of pearls”) are more suggestive for fungal etiology. Endogenous FE may start as flat choroidal lesions that progress to the vitreous cavity and lead to “puff ball” abscesses [3]. Vision loss can be mild in cases with peripheral vitreous lesions (“snowballs”, “snowbanks”) and severe in extensive vitreous and/or anterior chamber inflammation [6].

### 4.2. Imaging

B-scan ultrasonography is mandatory in eyes where there is no visualization of the posterior segment. While vitreous strands and membranes with reduced mobility are usual findings in endophthalmitis, the presence of a choroidal mass projecting into the vitreous (in the clinical context of an endophthalmitis) is suggestive for FE [3].

### 4.3. Laboratory Diagnosis

The ophthalmologist confronted with a possible diagnosis of endophthalmitis should perform an anterior chamber and/or vitreous tap before initiating treatment. It is known that anterior chamber tap has a lower diagnostic yield [9]. As often as possible, we prefer to sample undiluted vitreous during the beginning of a vitrectomy, using the technique described in the surgical management chapter. While the clinician may not initially suspect fungi as the etiology in a case of endophthalmitis, it is good practice to ask the laboratory to search for bacteria and fungi in the provided sample.

The usual workup includes direct microscopy (using stains as calcofluor white, Gram and Giemsa) and cultures on media such as blood agar, brain heart infusion, thioglicollate broth, potato dextrose agar and Sabouraud’s dextrose agar. It is necessary to incubate the media for 2 weeks before reporting a culture as negative [3].

Kehrmann et al. compared culture techniques in patients with suspected endophthalmitis and found that 100% of grown fungi were detected by blood culture bottles, while broth solution recovered 64% and solid media 46% of grown fungi [40]. It is also our usual practice to use blood culture bottles for immediate seeding of undiluted vitreous samples.

In cases of FE correlated with fungal keratitis, corneal scraping is also routinely recommended [6]. In a case series of *Fusarium* endophthalmitis, isolates were initially identified microscopically and the species subsequently confirmed by sequencing the elongation factor alpha (EFα) and internal transcribed spacers [41].

In endogenous FE it is recommended to perform blood cultures, even if they have a low diagnostic yield (9.2% to 25.6%) [9]. In order to maximize the yield, 3 consecutive blood samples should be taken during fever spikes and before systemic treatment [9,42].

The main advantage of culture techniques is that they are available in any hospital and the laboratories have extensive experience using them. Thus, microbiological culture remains the gold standard for the diagnosis of most intraocular infections. However, fungi may have a fastidious nature that makes them difficult to grow in culture (or, in some cases, unculturable). The rate of positive cultures in presumed FE has varied largely, between 30% and 70% [9].

Galactomannan (GM) is a cell wall carbohydrate that is mostly specific for *Aspergillus* species. While the manufacturer has only validated galactomannan detection in serum and bronchoalveolar lavage fluid, Dupont et al., have reported detection in a vitreous sample, suggesting that it might have a diagnostic role in cases with negative cultures and when PCR is not available [43].

β-d-glucan (BDG) is a major constituent of most fungal cell walls, including Candida and *Aspergillus* species. It can be detected in blood of patients with invasive fungal infections such as invasive candidiasis [44]. Chen et al., reported the testing of BDG in samples of intraocular fluids as a meaning of raising a high suspicion of FE (although the BDG concentrations in intraocular fluids of healthy individuals have not been established) [45]. Ammar et al., chose to define serum BDG ≥ 80 pg/mL as test positive and found a sensitivity of 66.7% for fungal chorioretinitis and 100% for endophthalmitis, while specificity was 74.4% [46]. A combination of PCR and BDG testing in patients with culture proven candidaemia and control patients revealed a sensitivity of 90% and a specificity of 79.5% [47].

The use of polimerase chain reaction (PCR) techniques has increased the yield of detection (up to 100%) [48] and has reduced the time necessary for etiological diagnosis. However the number of pathogens that can be simultaneously searched for is limited, due to differences in amplification efficiencies of different primer sets and to the limited number of fluorescent labels [49]. In a retrospective study on eyes suspected of endophthalmitis or infectious uveitis, cultures of aqueous humor or vitreous had 17% sensitivity, while PCR had 85% (remaining relatively inexpensive) [50]. It seems that PCR performed from aqueous humor and vitreous samples have similar diagnostic yields, which may ease the task of the first ophthalmologist who takes the patient into charge: it may be technically and logistically easier to collect a sample of aqueous humor before the initiation of treatment.

To date, the T2Candida panel (from T2 Biosystems) is the only commercial PCR assay platform with extensive clinical validation for the detection of *Candida* [51]. It detects the five major pathogenic *Candida* species: *C. albicans, C. tropicalis, C glabrata, C. krusei* and *C. parapsilosis.* We might observe that, with the advent of the COVID-19 pandemic, technology and expertise to perform PCR assays has known an unprecedented boom, and we hope to access it more frequently in the future for the diagnosis of other infectious diseases.

While authors were able to detect bacterial or fungal DNA in many cases of culture-negative endophthalmitis, the results have to be regarded with caution, and the clinician must balance the clinical and laboratory data while observing the response to therapy. Prior antimicrobial therapy is frequently incriminated for culture-negative endophthalmitis (while the microorganism may be dead, its DNA is still detectable). Other possible explanations are: the presence of fastidious microorganisms, scant (undetectable) bacterial pathogens or even true sterile endophthalmitis associated with antigenic response to a non-infectious pathogen [49].

While PCR techniques impose a limit on the number of pathogens that can be simultaneously searched for, next-generation sequencing (NGS) does not target specific species; it can detect all the different bacteria or fungi that are present in a sample, in one single assay. In a proof of concept study of 34 eyes with presumed infectious endophthalmitis, 44.1% were positive by microbial culture, while 82.3% were positive by the NGS technique. Among the culture negative endophthalmitis cases that showed presence of DNA of bacterial pathogens, 11 of 14 cases had polybacterial infections (4 had a bacterial and fungal infection) [49].

NGS is an emerging technique and it currently has a turnaround time of around 4–5 days. However, once it would become a routine test, Desmukh et al. predicted that a 48 h turnaround time would be feasible. The cost of NGS for metagenomics testing is somewhat comparable to the cost of current microbiological cultures, while promising in terms of reducing diagnostic time and (ultimately) hospitalization time [49].

While NGS, also termed high-throughput sequencing (HTS), has the potential to detect simultaneously and sequence virtually all the DNA seque nces present in a sample, it results in a large number of reads of both host and pathogen DNA. The detecting and interpreting of millions of sequences in order to identify the pathogen is highly challenging [52].

Targeted NGS uses a selective amplification of specific regions of interest inside the genome, prior to massive parallel sequencing. It provides easier downstream analysis and lower cost by allowing more samples to be tested in one run. In a study on vitreous samples from clinically presumed infectious endophthalmitis, Gandhi et al., used extraction and amplification of 16S RNA for the detection of bacteria and ITS 2 region for the detection of fungi. The rate of detection of fungal pathogens in culture-negative samples was 71.9%, again highlighting the prevalence of these pathogens in infectious endophthalmitis patients from South India [52].

The nanopore sequencer is a third-generation sequencing platform that identifies DNA from the change in electrical current resulting from a DNA strand being forced through a nanometer sized pore embedded in a membrane. In another very recent proof-of-concept study, Huang et al., have used nanopore targeted sequencing (NTS) in aqueous humor and vitreous fluid samples from presumed cases of infectious endophthalmitis [53]. NTS identified microorganisms in 94.4% of cases (half of which were culture-negative) [54].

A major critic for the use of genetic sequencing in the purpose of identifying pathogens is the lack of information about susceptibility to antimicrobial treatments. However, knowing the causative species can be helpful for the clinician, narrowing his choice of antimicrobials. There are exciting perspectives for culture-independent, molecular-based identification not only of pathogen fungi, but also of their antifungal resistance mechanisms [55]. To the date, there are no commercial PCR tests to detect mutations associated with antifungal resistance, but the latest developments in next-generation sequencing may allow in the future the detection of selected genes or regions associated with resistance [56,57]. Table 2 lists the main advantages and disadvantages of available diagnostic techniques.

In summary, the clinician that suspects a diagnosis of FE should perform a vitreous tap (or vitrectomy) and ask for microscopical examination and cultures for bacteria and fungi (we prefer initial seeding on blood culture bottles). For suspected endogenous endophthalmitis, blood cultures should be performed. If the endophthalmitis is keratitis-related, corneal scraping is also helpful. Searching for galactomannan and β-d-glucan in serum is fast and inexpensive. We should consult with the microbiologist and try to make the best use of the facilities available (perhaps PCR for organisms that are the most prevalent, considering the suspected mechanism of contamination).

### 4.4. Screening for Endogenous Endophthalmitis

As late as 2016, the Infectious Disease Society of America recommended a screening ophthalmological examination of all patients with candidemia [58]. However, recent studies have found that rates of ocular involvement in these patients were as low as 2.9% [59,60]. The American Academy of Ophthalmology has very recently stated that a routine ophthalmologic consultation after laboratory findings of systemic *Candida* septicemia appears to be a low-value practice. They have recommended that an ophthalmologic consultation should be performed in a patient with signs or symptoms suggestive of ocular infection, regardless of *Candida* septicemia [61].

## 5. Therapy

### 5.1. Medical Therapy

Amphotericin B is a polyene that binds surface sterols in the cell membrane of fungi, creating pores that alter the permeability, causing leakage of intracellular material and subsequent fungal cell death [62]. Azoles (like voriconazole) act by depleting ergosterol, a bioregulator of membrane integrity [6].

In suspected fungal endophthalmitis, initial treatment may be with intravenous voriconazole, loading dose 400 mg BID for two doses, then intravenous 300 mg/day (or oral 200 mg BID) AND intravitreal voriconazole 100 µg/0.1 mL. Voriconazole has excellent susceptibility to *Candida*, *Aspergillus* and *Fusarium* [63]. Monitor aspartate-aminotransferase (ASAT) and alanine-aminotransferase (ALAT) weekly for the first month. There are strong recommendations for the monitoring of voriconazole serum-levels [64]. When the laboratory has identified an etiologic agent, but its susceptibility to antifungals it yet to be determined, the clinician may use the suggestions of antifungal therapy presented in Table 3.

With the exception of purely chorioretinal fungal lesions (i.e., without vitritis), it is usually recommended to associate intravitreal therapy with the same antifungal used in intravenous or oral therapy, most frequently 100 µg/0.1 mL of voriconazole [63]. Another widely used intravitreal therapy is amphotericin B, 5–10 µg/0.1 mL. Intravitreal injections may be repeated after 72 h, depending on the clinical evolution [2]. Dave et al., reported that they have repeated intravitreal injections every 48 h for amphotericin B and every 24 h for voriconazole [24]. If the surgeon chooses to use silicone oil endotamponade at the conclusion of the vitrectomy, the dose of intravitreal anti-infection agents injected should be ¼-1/10 of the usual intravitreal dose [65].

The use of intravitreal corticosteroids in FE is controversial. An important concern is that corticosteroids may impair the efficacy of antifungals and interfere with the immunogenic response. A small retrospective study suggested that steroids may be beneficial in promoting faster clearance of inflammation in FE [66]. However, a review of the role of intravitreal corticosteroids in infectious endophthalmitis concluded that there is a lack of adequate experimental and human studies concerning steroids in FE [67]. Regarding oral steroids, we have found only one paper that advocated the use of oral prednisone, 1 mg/kg body weight in tapering doses [6]. When the laboratory results regarding antifungal susceptibility become available, the ophthalmologist may choose to change the therapy accordingly (however, we believe that a treatment that results in clinical improvement should not be changed based on laboratory findings). For susceptible *Candida* strains, fluconazole may be preferable to voriconazole because it is less hepatotoxic [63]. The usual antifungal doses are presented in Table 4. If effective, the systemic antifungal therapy should be continued for 3–6 weeks [63]

Fluconazole is preferred to voriconazole in children, because it is difficult to attain target voriconazole concentration and to monitor the serum levels. The loading dose is 12 mg/kg (intravenous or oral) fluconazole, followed by 6 mg/kg/day (or at 48 h for younger children) [68].

For azole-resistant *Candida* species, a combination of choice is intravenous amphotericin with oral flucytosine [58]. Liposomal amphotericin B may be less nephrotoxic and has higher vitreous penetration compared to amphotericin B deoxycholate [2]. *Candida* chorioretinitis without vitritis (which can occur in a setting of endogenous FE) can be treated with systemic therapy, without intravitreal injections [58]. Systemic treatment with echinocandins (micafungin, caspofungin, anidulafungin) may be effective in choroidal infiltrates with azole-resistant *Candida*, but are not effective in case of vitritis (due to low-moderate vitreous penetration) [4].

The Infectious Disease Society of America recommendations for the treatment of *Aspergillus* endophthalmitis are to combine oral or intravenous voriconazole with intravitreal voriconazole or intravitreal amphotericin B deoxycholate [69]. Azole-resistant species of *Aspergillus* may be treated with intravenous amphotericin B or intravenous anidulafungin. Itraconazole has a low minimum inhibitory concentration for *Aspergillus,* with better vitreous penetration than systemic amphotericin B (but the vitreous concentration remains a fraction of the serum level) [6].

For endophthalmitis caused by fungi other than *Candida* or *Aspergillus* there are very limited data regarding the immediate treatment, when the clinician has no information on the antifungals susceptibility. In *Fusarium* endophthalmitis there have been a few reports of successful treatment with systemic voriconazole (with or without amphotericin B) and intravitreal voriconazole [70,71].

#### 5.1.1. Alternative Antifungal Therapies

For cases of endophthalmitis caused by fungi that are resistant to usual antifungals, Relhan et al., have proposed (based on limited knowledge of retinal toxicity in animal studies): intravitreal miconazole 25 µg/0.1 mL, intravitreal caspofungin 50 µg/0.1 mL or intravitreal micafungin 25 µg/0.1 mL [72]. There are a few reports of successful treatment of FE using intravitreal injections of caspofungin [73,74,75,76].

Guest et al., studied the use of isavuconazole in treating *Aspergillus fumigatus* endophthalmitis in an exogenous mouse model of the disease and concluded that it was as effective after oral administration as it was after intravitreal administration [77]. Isavuconazole is a second-generation broad-spectrum triazole, noninferior to voriconazole for invasive aspergillosis and suitable for stepdown therapy in cases of invasive candidiasis [78]. Recently, oral isavuconazole was reportedly used in a case of *Candida* endophthalmitis unresponsive to fluconazole [79].

#### 5.1.2. Adverse Effects of Antifungal Treatment

Amphotericin B is nephrotoxic and may also be associated with variations of arterial pressure, fever or vomiting. However, liposomal amphotericin B has probably a lower rate of nephrotoxicity [2]. It should be used under the supervision of an internal medicine or infectious disease specialist [62].

Azole derivatives are hepatotoxic and should be use with caution in patients with pre-existing liver disease. Their risks include hepatic toxicity, cardia arrhythmias, fever and hypertension. Clinicians are advised to monitor aspartate-aminotransferase (ASAT) and alanine-aminotransferase (ALAT) weekly for the first month. There are strong recommendations for the monitoring of voriconazole serum-levels [64].

Caspofungin is a lipopeptide antifungal (from the echinocandins class). Risks associated include hepatotoxicity, Stevens-Johnson syndrome and toxic epidermal necrolysis [62].

### 5.2. Surgical Management

In the setting of acute postoperative endophthalmitis it can be difficult to immediately perform pars plana vitrectomy because of problems with operatory schedule. In those cases many ophthalmologists perform a vitreous needle tap and intravitreal injection, typically with antibiotics such as vancomycin and ceftazidime. However, most FE do not begin in an acute manner. Moreover, an inadvertent empirical antibiotic therapy does not preclude subsequent adequate identification of fungal etiological agents

There are no guidelines regarding the necessity and timing of pars plana vitrectomy in fungal endophthalmitis. There are several papers pleading for early and complete vitrectomy in bacterial endophthalmitis, and many surgeons have adopted this view [80,81,82]. Pars plana vitrectomy has the advantages of drastically reducing the load of intraocular microorganisms, providing a large sample for the microbiology laboratory and promoting the diffusion of antimicrobial drugs in the vitreous cavity.

Typically, after positioning the trocars, with the infusion cannula still closed, the surgeon will place the tip of the vitrector behind the lens (or in the center of the vitreous cavity). An assistant carefully aspires in a syringe coupled at the aspiration line of the vitrector, while the surgeon may choose to active cutting function. After the first drops are aspirated, the surgeon withdraws the vitrector and the assistant continues to draw all the drops from the tubing into the syringe, while the infusion is turned on. This technique usually provides at least 0.2 mL of undiluted vitreous that should be immediately sent for culture seeding.

Removal of “sticky” hypopyon and repeated subsequent irrigation of anterior chamber permits the visualization of vitreous cavity. After careful, patient removal of vitreous and repeated lavage of floating debris, the retina may become visible and the surgeon may attempt to remove fungal colonies from the retinal surface (bearing in mind that inflamed retina is friable and easily teared).

In post-cataract surgery FE, especially in cases that present recurrence after a vitrectomy, it is advisable to remove the intraocular lens and the capsular bag. Vinekar et al., have suggested that fungal spores sequestered in the capsular bag are responsible for the recurrences [83]. Relimpio-Lopez et al., pleaded for extreme surgical maneuvers that may help to salvage a globe with FE: “hot” penetrating keratoplasty (for perforated, infected cornea), iridectomy of infiltrated iris regions, endodiathermy or endophotocoagulation of chorioretinitis foci [84]. In a series of patients with posttraumatic FE, lensectomy was performed in all cases, together with pars plana vitrectomy [25].

A recent review of several series of patients with endogenous FE found that vitrectomy was performed in 24.2% to 56.9% of eyes [9]. In a retrospective cohort of 44 eyes with endogenous *Candida* endophthalmitis, Sallam et al., found that performing an early vitrectomy reduced significantly the risk of retinal detachment [85].

A more recent paper by Behera et al., studied the outcomes obtained in 66 consecutive patients with FE, divided in two groups based on the timing of vitrectomy. The patients in one group were subject to immediate vitrectomy, the others received a delayed diagnostic vitrectomy (after an average of 18 days). Both groups also received intravitreal antifungals. The authors concluded that the visual acuity improved significantly in the immediate vitrectomy group [86]. However, William et al., found that 42% of eyes with endogenous FE developed a retinal detachment. There was no association between the duration of symptoms and the development of retinal detachment [87].

In one retrospective study, nearly significant difference in final visual acuity (*p* = 0.06) was found in eyes that received intravitreal antifungals at the moment of the first vitrectomy, compared to those who received intravitreal antibiotics initially [25]. However, most ophthalmologists prefer to treat endophthalmitis with intravitreal antibiotics while awaiting microbiological diagnosis (simply because bacterial endophthalmitis is more prevalent that FE).

## 6. Prognosis

The visual prognosis in fungal endophthalmitis is poor, and exogenous FE has a worse prognosis than endogenous FE [2]. In a large retrospective series of 388 eyes with culture-positive endophthalmitis that were eviscerated, 35.3% were caused by fungi, suggesting a worse prognosis of FE relative to endophthalmitis of all causes [88]. The eyes with *Aspergillus* endophthalmitis present in a larger proportion with visual acuities inferior to hand movement (compared to eyes with other fungal etiologies) and this trend is maintained also when measuring the visual acuities at the end of the follow-up [3].

In a cohort of 342 patients with postoperative FE, 5.8% of eyes were eventually eviscerated and 13.7% of eyes had finally lost the light perception, while 30.1% of eyes gained a visual acuity ≥ 20/400 [3]. Sen et al., reported that 35.3% obtained a visual acuity of 6/60 or better, corneal involvement in addition to endophthalmitis and the presence of *Aspergillus terreus* being poor prognostic markers [89].

From 260 eyes with post-traumatic FE, 6.9% had no light perception at the end of the follow-up and 5% were eviscerated, but 31.1% had a final visual acuity of 20/400 or better [3]. Similar outcomes were found by Zhuang et al., in a cohort of patients from eastern China diagnosed with post-traumatic FE: 5.7% of eyes were finally enucleated, while 34.3% achieved a visual acuity ≥ 20/400 [25].

In the cohort of fungal endogenous endophthalmitis cases (128 patients), only one eye was eviscerated (0.8%). In 17.9% the final visual acuity was no light perception and in 10.9% of eyes it was ≥20/400. All patients were treated with vitreous surgery and at least one intravitreal injection of antifungals [3]. In a review of several series of endogenous FE cases (where primary vitrectomies were performed in 24.2% to 56.9% of eyes), intravitreal antifungals were administered in 54% to 100% of eyes and were repeated in 33% to 50%. All authors have reported high rates of anatomical success (75% to 100%). However, functional success (defined as final visual acuity ≥ 20/400) was reported in only 33% (in mold infections) to 56% of eyes (in yeast infections). Patients with mold infections had worse visual acuities, both at the presentation and during the follow-up [9].

In a smaller series of patients with FE from North China, of which almost half were diagnosed with *Fusarium* endophthalmitis, 56.4% obtained final visual acuity of 20/400 or better [27].

In a retrospective study of eyes with *Aspergillus* endophtalmitis, 34% eyes obtained a final visual acuity ≥ counting fingers (21.9% final VA ≥ 20/400). The factors associated with better visual outcomes were: presenting vision greater than hand motions, absence of corneal infiltrate, early vitrectomy and the use of intravitreal voriconazole (as compared to intravitreal amphotericin B) [24].

## 7. Discussion

One challenge for the ophthalmologist who has to manage a case of endophthalmitis is to bear in mind the potential fungal etiology. Patients that present the first symptoms weeks after an ocular surgery or trauma (or have risk factors for endogenous endophthalmitis), a clinical examination that reveals focal vitreous opacities or choroidal lesions, the prevalence of fungal infections in a certain geographical area are factors that should increase the level of suspicion, but fungi have to be taken into account as potential etiology in all cases.

FE has a particularly high prevalence in tropical regions, and today the best source of information regarding this pathology is provided by authors from Asia. Molds (and in particularly *Aspergillus*) are a common etiology for FE in southern Asia [3,24,25,90].

While benefitting from the experience published by many authors about this particular etiology, clinicians from other geographical areas should be aware that other fungi may be more prevalent in their area and that the antifungal susceptibilities may also be different. Studies published by authors from other countries generally found a higher prevalence of yeast infections.

Many presumed FE cases remain culture-negative, and there is a lot of interest in the use of PCR assays to obtain etiological diagnosis. Emerging techniques such as next-generation sequencing (also termed high-throughput sequencing) and nanopore targeted sequencing come with the promise of detecting virtually all DNA sequences present in a sample, and we hope that they will become more available (and the cost will decrease). In the meantime, clinicians should remember the diagnostic potential of less expensive tests such as those for galactomannan and β-d-glucan.

In endogenous FE, more than ¼ of patients had a bilateral involvement. Blood samples have a low diagnostic yield. Vitrectomy was performed in a low proportion of cases (sometimes patients with a systemic infection cannot withstand a surgery due to the severity of general status). Contrary to exogenous FE, an intravitreal injection was not performed in all cases, some authors preferring to rely solely on intravenous therapy. Anatomical success rate was high, but that of functional success was low (especially in cases caused by molds) [3,9].

Regarding all types of FE, there is a concern about new etiologic agents being identified and also about the prevalence of antifungal resistance, while newer antifungals are yet to be tested in this particular pathology (or have low efficiency because of low penetration in the vitreous, as is the case of echinocandins). Several authors reported largely similar outcomes in exogenous FE, with about 5% of eyes that were eventually enucleated, while visual acuities ≥20/400 were obtained in about 1/3 of cases [3,25,89].

The recent publishing of large case series of patients with fungal endophthalmitis is providing the ophthalmologists with a level of knowledge that is unprecedented in this area, while the development of diagnostic techniques gives us hope to have an etiologic diagnosis of eye infections in almost the totality of cases, and perhaps also information about the presence of DNA sequences that codify antifungal resistance.

## Figures and Tables

**Table 1 diagnostics-12-00679-t001:** The prevalence of fungal endophthalmitis.

	Country	Number of Studied Cases	Percentage of Fungal Etiology
Das et al. [3]	India	3830 cases (culture proven endophthalmitis)	19.1
Schimel et al. [5]	USA	448 cases	15.8
Long et al. [7]	China	347 cases (culture proven post-traumatic endophthalmitis)	16.8
Yang et al. [10]	China	151 cases (culture-proven endophthalmitis associated with intraocular foreign bodies-IOFB)	8
Yang et al. [10]	China	256 cases (culture-proven post-traumatic endophthalmitis)	15.6
Dave et al. [11]	India	117 patients endogenous endophthalmitis (EE)	15
Regan et al. [12]	USA	35 patients with EE	
Pillai et al. [13]	India	34 patients with EE	50
Cho et al. [14]	USA	60 patients with EE	34.3
	Korea	48 patients with EE	16.4
Kuo et al. [15]	Taiwan	31 patients with EE	8
Kuo et al. [15]	Taiwan	25 patients with EE and chronic dialysis	4
Silpa-archa et al. [16]	Thailand	36 patients with EE	7.3
Modjtahedi et al. [17]	USA	30 patients with intravenous drug abuse-related EE	59
Maitray et al. [18]	India	53 pediatric patients with EE	6

**Table 2 diagnostics-12-00679-t002:** Advantages and disadvantages of currently available diagnostic techniques.

Technique	Advantages	Disadvantages
**Microbiological culture**	- microscopy can rapidly orient diagnosis (yeasts versus molds) - widely available, extensive experience - relatively low cost - tests for susceptibility to antifungals	- time consuming (up to 2 weeks) - relatively low diagnostic yield
**Detection of fungal cell wall constituents (galactomannan, β-d-glucan)**	- widely available - relatively low cost - fast results - reportedly present in vitreous (but not a standardized technique)	- validated only for detection in blood (implying invasive fungal infection)
**Polymerase chain reaction (PCR)**	- fast results (several hours) - increased yield (compared to cultures) - extensive clinical validation for the detection of *Candida* species - apparently similar sensitivity from aqueous humor and vitreous samples - positive even if antimicrobial treatment has been started - ever more widely available	- limited number of fluorescent labels (hence, of pathogens that can be simultaneously searched for) - no information about antifungal susceptibility
**Next generation sequencing (NGS)**	- can detect all the different organisms present in a sample - possible future detection of genes associated with antifungal resistance	- not widely available today - time consuming (several days)

**Table 3 diagnostics-12-00679-t003:** Suggested initial therapy in fungal endophthalmitis (FE) cases were an etiologic agent has been identified, but antifungal susceptibility is yet unknown.

**Etiology**	**Suggested Initial Therapy**
*Candida*	Intravenous/oral fluconazole OR
	Intravenous voriconazole OR
	Intravenous micafungin (only in chorioretinitis without vitritis) *
*Aspergillus*	Intravenous voriconazole
*Fusarium*	Intravenous voriconazole OR
	Oral posaconazole
Other etiologies	Intravenous voriconazole OR
	Intravenous amphotericin B associated with oral fluconazole

* Intravenous echinocandins (micafungin, anidulafungin, caspofungin) may be effective in chorioretinal fungal infiltrates, but have low-moderate vitreous penetration.

**Table 4 diagnostics-12-00679-t004:** Antifungal treatment regimens for FE.

**Antifungal**	**Recommended Regimen**
Voriconazole	i.v. 400 mg BID, two doses, then 300 mg/day (or oral 200 mg BID)
Fluconazole	i.v. /oral 800 mg loading dose, then 400–800 mg/day [2]
Liposomal amphotericin B	i.v. 3–5 mg/kg/day
Flucytosine	oral 25 mg/kg, QID
Ketoconazole	oral 200 mg BID [3]
Posaconazole	oral 200 mg QID
Itraconazole	oral 100–200 mg BID [3,6]
Micafungin	i.v. 100–300 mg/day
Anidulafungin	i.v. 100–200 mg/day
Caspofungin *	i.v. 70 mg loading dose, then 50 mg/day [62]

* Only in cases without vitritis.

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
