# Peer review of "The Diagnosis and Treatment of Fungal Endophthalmitis: An Update"

_diagnostics, 2022, doi:10.3390/diagnostics12030679_

Round 1
Reviewer 1 Report
Excellent update / review of fungal endophthalmitis. The information on new diagnostic methodologies (PCR etc. ) is particularly useful.
Suggestions to authors:
Line 16 "germ" is technically correct but seems passe. Suggest changing to "organism".
Section 2. lines 52-71 if feasible, please present epidemiology as a table.
4.3 is particularly interesting. Can the authors give guidance on the preferred approach to seeking a rapid and cost-effective path to diagnosis? How about a box/decision tree?
Lines 227-228 please state where the cost is - EU, USA or average worldwide?
Line 362 change "install" to ‘’begin”
Author Response
We would like to take the opportunity to thank you again for the time dedicated to our paper. We have done our best to answer to all the concerns expressed by the reviewers, as detailed below:
Reviewer no. 1:
Suggestions to authors:
Line 16 "germ" is technically correct but seems passe. Suggest changing to "organism". We have performed the suggested change.
Section 2. lines 52-71 if feasible, please present epidemiology as a table. We have included the following table (however, the formatting changed when we have put it in the provided template).
|
Country |
Number of studied cases |
Percentage of fungal etiology |
Das et al. [3] |
India |
3830 cases (culture proven endophthalmitis) |
19.1 |
|
|
|
|
Schimel et al. [5] |
USA |
448 cases |
15.8 |
Long et al [7] |
China |
347 cases (culture proven post-traumatic endophthalmitis) |
16.8 |
Yang et al [9] |
China |
151 cases (culture-proven endophthalmitis associated with IOFB) |
8 |
Yang et al [9] |
China |
256 cases (culture-proven post-traumatic endophthalmitis without IOFB) |
15.6 |
Dave et al. [11] |
India |
117 patients endogenous endophthalmitis (EE) |
15 |
Regan et al. [12] |
USA |
35 patients with EE |
|
Pillai et al. [13] |
India |
34 patients with EE |
50 |
Cho et al. [14] |
USA |
60 patients with EE |
34.3 |
|
Korea |
48 patients with EE |
16.4 |
Kuo et al. [15] |
Taiwan |
31 patients with EE |
8 |
Kuo et al. [15] |
Taiwan |
25 patients with EE and chronic dialysis |
4 |
Silpa-archa et al. [16] |
Thailand |
36 patients with EE |
7.3 |
Modjtahedi et al. [17] |
USA |
30 patients with intravenous drug abuse-related EE |
59 |
Maitray et al. [19] |
India |
53 pediatric patients with EE |
6 |
4.3 is particularly interesting. Can the authors give guidance on the preferred approach to seeking a rapid and cost-effective path to diagnosis? How about a box/decision tree?
We have received conflicting advice from the reviewers concerning chapter 4.3. Unfortunately, we were not able to provide a box/decision tree for the diagnosis of fungal endophthalmitis. Even in our university hospital in Iasi, the possibilities of diagnosis would be restricted to cultures and tests for fungal cells constituents (we do not currently have diagnostic PCR kits for fungi). We have decided to provide a table with the advantages and disadvantages of each diagnostic method, and we have summarized our suggested approach at the end of the chapter:
In summary, the clinician that suspects a diagnosis of FE should perform a vitreous tap (or vitrectomy) and ask for microscopical examination and cultures for bacteria and fungi (we prefer initial seeding on blood culture bottles). For suspected endogenous endophthalmitis, blood cultures should be performed. If the endophthalmitis is keratitis-related, corneal scraping is also helpful. Searching for galactomannan and β-D-Glucan in serum is fast and inexpensive. We should consult with the microbiologist and try to make the best use of the facilities available (perhaps PCR for organisms that are the most prevalent, considering the suspected mechanism of contamination).
Lines 227-228 please state where the cost is - EU, USA or average worldwide?
The costs were taken from a paper written by an Indian author [49]. We do not have informations about the average cost in EU or the USA, so we have opted not to give a certain number:
The cost of NGS for metagenomics testing is somewhat comparable to the cost of current microbiological cultures, while promising in terms of reducing diagnostic time and –ultimately- hospitalization time [49].
Line 362 change "install" to ‘’begin”
We have performed the suggested change
With respect,
Ciprian Danielescu , Horia Tudor Stanca , Raluca-Eugenia Iorga , Diana-Maria Darabus and Vasile Potop
Reviewer 2 Report
The present manuscript is a thorough, yet concise well-written review of the diagnosis and treatment of fungal endophthalmitis. Fungal endophthalmitis is an important disease process in the field of ophthalmology and an updated overview of the management is important for the clinical and scientific communities. I really do not have many comments to make for the paper other than minor grammatical corrections for the manuscript, but otherwise feel that the manuscript is fit for publication as is.
Author Response
We would like to take the opportunity to thank you again for the time dedicated to our paper. We have done our best to answer to all the concerns expressed by the reviewers, as detailed below:
Reviewer no 2: The present manuscript is a thorough, yet concise well-written review of the diagnosis and treatment of fungal endophthalmitis. Fungal endophthalmitis is an important disease process in the field of ophthalmology and an updated overview of the management is important for the clinical and scientific communities. I really do not have many comments to make for the paper other than minor grammatical corrections for the manuscript, but otherwise feel that the manuscript is fit for publication as is.
The manuscript has been revised by a professor of English language agreed by our University of Medicine in Iasi.
With respect,
Ciprian Danielescu , Horia Tudor Stanca , Raluca-Eugenia Iorga , Diana-Maria Darabus and Vasile Potop
Reviewer 3 Report
In the current review, the authors reviewed the fungal endophthalmitis diagnosis and treatments. The authors summarized the recent advancement in the NGS and challenges. The various available treatments are also described. Overall, the review has addressed the important ocular infection but needs to address the following shortcomings.
1) Authors have described the antifungal therapies but anti-inflammatory therapies information is lacking. In line 296 (only one line) use of intravitreal dexamethasone is mentioned. Authors can expand and provide the advantage as well as disadvantages of therapy.
2) The organization of the review needs extensive editing. In many places, the paragraphs are of one or two lines. The rationale of the Result section in the review is not clear.
3) In the diagnosis of fungal ocular infection authors should provide the importance and drawbacks of fungal culture methods. Authors can summarize the diagnostic methods in a table, describing the advantage and disadvantages of each method.
Author Response
We would like to take the opportunity to thank you again for the time dedicated to our paper. We have done our best to answer to all the concerns expressed by the reviewers, as detailed below:
Reviewer no.3: In the current review, the authors reviewed the fungal endophthalmitis diagnosis and treatments. The authors summarized the recent advancement in the NGS and challenges. The various available treatments are also described. Overall, the review has addressed the important ocular infection but needs to address the following shortcomings.
- Authors have described the antifungal therapies but anti-inflammatory therapies information is lacking. In line 296 (only one line) use of intravitreal dexamethasone is mentioned. Authors can expand and provide the advantage as well as disadvantages of therapy.
We have added a paragraph, as follows:
The use of intravitreal corticosteroids in FE is controversial. An important concern is that corticosteroids may impair the efficacy of antifungals and interfere with the immunogenic response. A small retrospective study suggested that steroids may be beneficial in promoting faster clearance of inflammation in FE [89]. However, a review of the role of intravitreal corticosteroids in infectious endophthalmitis concluded that there is a lack of adequate experimental and human studies concerning steroids in FE [90]. Regarding oral steroids, we have found only one paper that advocated the use of oral prednisone, 1 mg/ kg body weight in tapering doses [6]
- The organization of the review needs extensive editing. In many places, the paragraphs are of one or two lines. The rationale of the Result section in the review is not clear.
We acknowledge that the title “Results” may be misleading and we have changed it into “Prognosis”. We believe that this chapter should provide the clinician with data from several papers regarding the prognosis of FE, thus helping to obtain the informed consent from the patient.
We have re-edited the entire paper and changed the organization of several paragraphs, hoping for a better, logical narration.
- In the diagnosis of fungal ocular infection authors should provide the importance and drawbacks of fungal culture methods. Authors can summarize the diagnostic methods in a table, describing the advantage and disadvantages of each method.
We have added Table 2. The advantages and disadvantages of diagnostic methods
Technique |
Advantages |
Disadvantages |
Microbiological culture |
- microscopy can rapidly orient diagnosis (yeasts versus molds) -widely available, extensive experience -relatively low cost - tests for susceptibility to antifungals |
-time consuming (up to 2 weeks) -relatively low diagnostic yield
|
Detection of fungal cell wall constituents |
-widely available -relatively low cost -fast results -reportedly present in vitreous (but not a standardized technique) |
-validated only for detection in blood (implying invasive fungal infection) |
Polymerase chain reaction (PCR) |
-fast results (several hours) -increased yield (compared to cultures) -apparently similar sensitivity from aqueous humor and vitreous samples -positive even if antimicrobial treatment has been started - ever more widely available |
-limited number of fluorescent labels (hence, of pathogens that can be simultaneously searched for) -no information about antifungal susceptibility |
Next generation sequencing (NGS) |
-can detect all the different organisms present in a sample -possible future detection of genes associated with antifungal resistance |
-not widely available today - time consuming (several days)
|
With respect,
Ciprian Danielescu , Horia Tudor Stanca , Raluca-Eugenia Iorga , Diana-Maria Darabus and Vasile Potop